# The Role of Dysbiotic Oral Microbiota in Cardiometabolic Diseases: A Narrative Review

**DOI:** 10.3390/diagnostics13203184

**Published:** 2023-10-12

**Authors:** Sylvie Lê, Chiara Cecchin-Albertoni, Charlotte Thomas, Philippe Kemoun, Matthieu Minty, Vincent Blasco-Baque

**Affiliations:** 1Département Dentaire, Université Paul Sabatier III (UPS), 3 Chemin des Maraîchers, CEDEX 9, 31062 Toulouse, France; sylvie.le@inserm.fr (S.L.); chiara.cecchin-albertoni@inserm.fr (C.C.-A.); charlotte.thomas@inserm.fr (C.T.); philippe.kemoun@inserm.fr (P.K.); matthieu.minty@inserm.fr (M.M.); 2Service d’Odontologie Toulouse, CHU Toulouse, 3 Chemin des Maraîchers, CEDEX 9, 31062 Toulouse, France; 3Team InCOMM/Intestine ClinicOmics Metabolism & Microbiota, Institut des Maladies Métaboliques et Cardiovasculaires (I2MC)—UMR1297 Inserm, Université Paul Sabatier, 1 Avenue Jean Poulhes, 31432 Toulouse, France; 4RESTORE Research Center, CNRS, EFS, ENVT, Batiment INCERE, INSERM, Université de Toulouse, 4 bis Avenue Hubert Curien, 31100 Toulouse, France

**Keywords:** oral microbiota, cardiometabolic diseases, cardiovascular diseases, microbiome, prebiotics/probiotics, translocation, periodontitis

## Abstract

Over the past decade, there have been significant advancements in the high-flow analysis of “omics,” shedding light on the relationship between the microbiota and the host. However, the full recognition of this relationship and its implications in cardiometabolic diseases are still underway, despite advancements in understanding the pathophysiology of these conditions. Cardiometabolic diseases, which include a range of conditions from insulin resistance to cardiovascular disease and type 2 diabetes, continue to be the leading cause of mortality worldwide, with a persistently high morbidity rate. While the link between the intestinal microbiota and cardiometabolic risks has been extensively explored, the role of the oral microbiota, the second-largest microbiota in the human body, and specifically the dysbiosis of this microbiota in causing these complications, remains incompletely defined. This review aims to examine the association between the oral microbiota and cardiometabolic diseases, focusing on the dysbiosis of the oral microbiota, particularly in periodontal disease. Additionally, we will dive into the mechanistic aspects of this dysbiosis that contribute to the development of these complications. Finally, we will discuss potential prevention and treatment strategies, including the use of prebiotics, probiotics, and other interventions.

## 1. Introduction

Cardiovascular diseases (CVDs) represent a complex and significant public health challenge, standing as one of the primary causes of morbidity and mortality in developed nations. Understanding the pathophysiology of CVDs is crucial for both basic and clinical research. It is widely recognized that these diseases, often associated with atherosclerosis, develop gradually under the influence of genetic predispositions and environmental factors. Therefore, they are multifactorial in nature. While age, smoking, dyslipidemia, arterial hypertension, insulin resistance and obesity have been identified as key cardiovascular risk factors, these factors alone do not fully explain the individual variability in the risk of cardiovascular morbidity and mortality. Consequently, there is a growing need to identify new explanatory risk factors. The emerging role of the immuno-inflammatory process as a common link among these risk factors contributing to the pathogenesis of cardiovascular diseases has become increasingly evident. Despite efforts to raise public awareness and improve lifestyle and dietary habits, thus positively impacting the gut microbiota, the global prevalence of cardiovascular diseases continues to rise. This suggests the existence of other risk factors in these diseases, such as the oral microbiota. Indeed, the oral cavity is one of the largest known reservoirs of bacteria within the human body, making it a vital component of the overall microbiota [1]. With numerous ecological niches present in both soft and hard tissues, the oral microbiota plays a crucial role in the local and systemic health of an individual. 

Recently, extensive research has investigated the potential connection between COVID-19 and periodontitis. It is notable that the bacteria implicated in COVID-19 infection seem to share similarities with those found in the oral microbiota. This observation suggests the question of “how periodontal issues might contribute to the progression of the infection”. The tissues impacted by periodontitis create access for bacterial or viral pathogens, such as SARS-CoV-2. This is due to clinical signs characterized by extensive ulcerated regions, establishing periodontitis as a significant point of entry for these pathogens. Furthermore, periodontitis worsens systemic inflammation, leading to the release of pro-inflammatory cytokines and tissue-damaging agents into the circulatory system, such as IL-1β, IL-10, IL-17, Th17, IFN-gamma, GM-CSF, G-CSF, IL-8, TNF-α, and MCP1 [2]. Furthermore, both periodontal disease and COVID-19 appear to incite and/or hinder various cardiometabolic issues, including cardiovascular disease, type 2 diabetes, metabolic syndrome, dyslipidemia, insulin resistance, and obesity [3]. These findings emphasize the necessity for a comprehensive exploration of this intricate relationship between COVID-19 and oral health, particularly regarding periodontitis, and the consequences in cardiometabolic diseases (CMDs) by contributing to the aggravation of inflammatory [3].

Through a narrative review approach, which involves synthesizing and analyzing the existing literature to offer a comprehensive and unified overview of the topic, this article aims to analyze the evidence for the association between the oral microbiota and cardiometabolic risk factors such as obesity, diabetes, and hypertension, leading to various cardiovascular diseases including atherosclerotic cardiovascular diseases, heart failure, and infective endocarditis. We will investigate the molecular mechanisms through which oral dysbiosis and associated oral pathologies contribute to the risk of developing cardiovascular diseases. In the final part, this review will discuss future research directions, potential prevention strategies, and their clinical and therapeutic applications.

## 2. Methods

### Search Strategy

In conducting this narrative review, the authors utilized various databases, including PubMed, Web of Science, Google Scholar, and the Cochrane Library, without imposing restrictions based on country or publication date. Search terms included the following: (oral microbiota OR oral dysbiosis OR salivary microbiota OR periodontal microbiota) AND (cardiometabolic diseases OR cardiometabolic risk OR cardiovascular diseases OR CMDs OR CVDs). Additionally, relevant articles were identified through both backward searching, which involved reviewing the references of located articles, and forward searching, which entailed finding newer articles that included the original cited papers. Bibliographic searches were iteratively performed to ensure that the most recent advancements in the field were taken into account.

## 3. Oral Microbiota

### 3.1. Generalities

The human microbiota consists of a large number of microorganisms coexisting in various sites of the human body. While the term “microbiota” describes the microbial communities living in a specific environment (e.g., the gut), the term “microbiome” refers to the genome from all the microorganisms in an ecological niche [4]. Different microbial communities can be found in several organs, like the gut, oral cavity, lung, vagina, or skin [5]. In healthy conditions, microbial communities establish a symbiotic relationship with the host and contribute to maintaining physiological homeostasis (eubiosis), which is crucial for overall well-being. On the other hand, an imbalance in microbial communities can disrupt the equilibrium between the host and its microbiota (dysbiosis). The term dysbiosis refers to an imbalance in the microbial community, where harmful bacteria may become overabundant and be associated with several diseases such as obesity, type 2 diabetes mellitus, inflammatory bowel disease, and cardiovascular diseases [6,7].

The oral cavity harbors the second-most-significant microbial reservoir in the body, after the gut [8]. The oral microbiota consists of almost 700 kinds of microorganisms, including bacteria, viruses, protozoa, fungi, and phages [9]. Its acquisition begins immediately after birth through a vertical transmission between mother and child (contact with mother’s skin, vagina, and oral microbiota), and initially consists of facultative anaerobic bacteria of the genus *Streptococcus*. Later, strict anaerobic bacteria, such as the genus *Veillonellae* (Firmicutes) and the Phylum Fusobacteria, colonize the mouth.

The oral cavity is a complex ecosystem with several niches, including the oral epithelium, the tongue, the supra-gingival dental surfaces and the sub-gingival space [10]. As a result, microorganisms in the mouth can exist in different states, either as planktonic cells in the liquid environment or as biofilm attached to the tooth surface (which represent the only non-desquamating surface in the human body) [11]. The microbiota composition in all sites shares overall similarities but with small scale differences. In general, the major bacteria present in oral microbiota are Firmicutes, Proteobacteria, Bacteroidetes, Actinobacteria, and Fusobacteria [5]. The symbiotic relationship between the host and the oral microbiota is constantly challenged by internal and external ecological changes. The composition of the oral microflora depends on life events, such as food diversification, tooth eruption, hormonal changes (e.g., puberty, menstrual period, pregnancy), medications (including antibiotics), and aging [12]. While the oral microbiota demonstrates resilience to minor ecological disturbances, prolonged perturbations can disrupt the balance and lead to dysbiosis, which can be associated with oral and systemic diseases such as periodontitis, dental caries, or cardiometabolic diseases [11,12]. Understanding the composition and dynamics of the oral microbiota is therefore crucial for maintaining oral and systemic health. Saliva and periodontium represent two key sites for oral microbiota.

### 3.2. The Salivary Microbiota

Saliva is a unique fluid, produced by several salivary glands, that covers the surfaces of the oral cavity and plays a crucial role in various physiological processes such as chewing, swallowing, and speaking [11]. It also contains important biological components, including proteins (such as mucins and glycoproteins) and enzymes, which play a critical role in providing nutrition, controlling the growth of potentially harmful microorganisms and promoting oral health [13].

The salivary microbiota consists of microorganisms shed from the environmental surfaces within the oral cavity (including teeth, gingival sulcus, cheeks, hard and soft palates, gums, tongue, tonsils, etc.), and is unique and specific to each individual [13]. It is relatively stable over time but is influenced by factors such as nutrition and lifestyle [6]. The salivary microbiota comprises 500 to 700 bacterial species [14], and it also includes other microorganisms such as archaebacteria, protozoa (e.g., *Entamoeba gingivalis* and *Trichomonas tenax*), and up to 85 species of fungi (including Candida, Cladosporium, Aureobasidium, Saccharomycetales, Aspergillus, Fusarium, and Cryptococcus) [15]. Viruses, mainly bacteriophages, can also be present in the salivary microbiota (e.g., Mumps virus, HIV-1, and the SARS-CoV-2). Analyzing the salivary microbiota allows for the prognosis of tooth decay, as the presence of certain bacteria such as *Streptococcus mutans*, *Rothia*, *Fusobacterium*, *Prevotella*, *Leptotrichia*, and *Capnocytophaga* can indicate a carious state [16]. Additionally, changes in the composition of the salivary microbiota are characteristic of other oral and systemic conditions, like periodontal diseases, type 2 diabetes, obesity, or atherosclerosis [11]. Saliva plays a crucial role in shaping the oral microbiota, and the analysis of the salivary microbiota can provide insights into health conditions.

### 3.3. The Periodontal Microbiota

The periodontium represents different tooth supporting structures and consists of the gingiva (or gum), cementum, periodontal ligament, and alveolar bone. The gingiva (superficial periodontium) is a resilient oral mucosa with a keratinized epithelium and a fibrous connective tissue rich in collagen fibers that protects the deep periodontium; it also contains defense cells that play a role in the immune reaction against oral threat. The gingiva is attached to the tooth by a permeable epithelial–conjunctive attachment.

The sulcus, the groove between the tooth and the gum line, contains a fluid with microorganisms, cell debris, and electrolytes. Evacuation of this fluid helps to protect the deep periodontium. The deep periodontium comprises the periodontal ligament, cementum, and alveolar bone, forming the alveolar attachment system that provides stability and cushioning for the tooth [17].

The periodontal microbiota, which emerges during the eruption of the first teeth, consists of a combination of planktonic and biofilm bacteria embedded in an exo-polysaccharide matrix [9] that develops in the sulcus and adheres to the root surface. Periodontal health is associated with a predominance of Gram-positive cocci and rods, such as *Actinomyces naeslundi*, which serve as primary colonizers on the root surface and support the development of dental plaque biofilm. The symbiotic periodontal microbiota coexists with the host in a state of homeostasis. The host’s immune system helps to control microbial colonization and maintain periodontal health [18]. However, in susceptible individuals, certain factors such as genetics, epigenetics, aging, lifestyle, and environmental factors can disrupt the host–microbiota homeostasis, leading to dysbiosis and to a shift toward a pathogenic microbiota [19]. Dysbiosis is, hence, the consequence of a change in the dominant species rather than a de novo bacterial colonization [20]. In periodontal diseases, the dysbiotic periodontal microbiota is essentially composed of strict anaerobic Gram-negative bacteria (*Treponema denticola*, *Porphyromonas gingivalis*, and *Tannerella forsythia*) [21,22]. As the dysbiosis and inflammation progress, the destructive immuno-inflammatory response leads to the breakdown of the junctional epithelium and to the infiltration of inflammatory cells into the deep periodontium, causing the recruitment of immune cells and the generation of an inflammation amplification loop, which leads to the destruction of periodontal tissues and tooth loss (so-called periodontitis). The prevalence and severity of periodontitis increases with age. Almost 50% of the world’s adult population have periodontal disease, and severe periodontitis affects around 19% of the global adult population, representing more than 1 billion cases worldwide [23]. Periodontitis is not limited to its local effects in the oral cavity, but has also been associated with various systemic conditions, including diabetes, adverse pregnancy outcomes, and cardiovascular diseases [24,25]. Understanding these associations can help improve patient care and overall health outcomes.

### 3.4. Summary

Table 1 summarizes the characteristics and differences between the salivary microbiota and the periodontal microbiota. From this table, we will focus on the species most present in these microbiotas and summarize their involvement in CMDs (Table 2).

Indeed, Streptococcaceae, a Gram-positive bacteria family, is involved in cardiometabolic risk with an increase in high blood pressure (HBP) and a low HDL concentration [26]. *Streptococcus mutans* is also enriched in type 2 diabetes (T2D) microbiota [27]. *Staphylococcus aureus* is associated with cholesterol and negatively associated with triglycerides, that characterizing the cardiometabolic alterations in children [30]. *Fusobacterium nucleatum* accelerates atherosclerosis via a macrophage pro-inflammatory response [31]. The association of Tannerellaceae with obesity and hypertension has been fully described [32]. In contrast, Neisseriaceae, a Gram-negative bacteria family, is associated with a decrease in HBP and an improvement in vascular function in patients with hypercholesterolemia [28]. *Prevotella copri*, belonging to the Prevotellaceae family, is associated with an improvement in glucose tolerance [29].

## 4. Cardiovascular Diseases

Cardiovascular diseases (CVDs) are complex multifactorial conditions (including, i.e., coronary heart disease, cerebrovascular disease, peripheral arterial disease) and represent one of the leading causes of morbidity and mortality in industrialized countries [33]. According to the Global Burden of Disease Study 2019, cardiovascular diseases are responsible for approximately 18.6 million deaths globally, accounting for 32.0% of all deaths [34]. The burden of cardiovascular diseases varies across different regions of the world. While high-income countries have traditionally experienced higher rates of cardiovascular diseases, low- and middle-income countries are now also facing an increasing prevalence [35]. CVDs have long been seen as a condition primarily affecting men. Although the age-specific rates of CVDs are higher in men than women in most age groups, the actual lifetime risk of CVDs is similar for women and men [36]. CVDs develop gradually under the influence of environmental factors and genetic predispositions. While age, smoking, dyslipidemia, arterial hypertension, insulin resistance, and overweight are established cardiovascular risk factors, they do not fully explain the variability in individual risk [37]. Despite efforts to reduce risk factors and improve lifestyle and diet, the global prevalence of cardiovascular diseases continues to increase, suggesting the presence of other contributing factors.

The immuno-inflammatory process has emerged as a common factor among cardiovascular risk factors and plays a significant role in the development of cardiovascular pathologies [38]. The oral microbiota, which is one of the largest reservoirs of known bacteria in the human body, is increasingly being studied as a potential risk factor for cardiovascular diseases. The oral cavity contains numerous ecological niches on soft and hard tissues, highlighting the important role of the oral microbiota in the overall health relationship [1]. Understanding the connections between the oral microbiota and cardiovascular diseases can provide new insights into the etiology and management of these conditions.

### 4.1. Epidemiologic Evidence of the Association between Periodontitis and Cardiovascular Diseases

The correlation between periodontitis and myocardial infarction was first reported by Mattila et al., marking the beginning of studies exploring the link between periodontitis and cardiovascular disease [39]. Subsequent epidemiological studies have consistently shown a significant association between periodontitis and CVDs [40], although the underlying causal pathophysiological relationship remains to be fully understood. It has been observed that periodontitis can contribute to endothelial dysfunction and increased arterial calcification scores, suggesting its potential role in cardiovascular health [41]. Additionally, a positive association between periodontitis and cerebrovascular disease and the risk of stroke has been documented. Campanella et al. highlighted the higher prevalence of periodontitis in patients with stroke compared to controls [42]. A recent meta-analysis concluded that individuals with periodontitis have twice the risk of suffering from a stroke and are at a higher risk of experiencing stroke compared to those without periodontal disease [43]. Furthermore, periodontitis has been positively associated with an increased risk of heart failure [44] and coronary heart disease (CHD) [45]. Studies have shown a significant association between severe periodontitis and an increased incidence of CHD, independent of other cardiovascular risk factors [46]. The relationship between periodontitis and the composite of coronary artery disease and stroke has also been examined; patients with severe periodontal disease are more likely to have CVDs compared to those with mild or subclinical periodontitis [47]. Limited but consistent evidence suggests that individuals with periodontitis have a higher prevalence and incidence of peripheral artery disease (PAD) compared to those without periodontitis, and this association is independent of other risk factors [48]. PAD patients also present a higher risk of developing periodontitis compared to non-PAD individuals [49]. Currently, there is no strong evidence highlighting cardiovascular disease as a risk factor for the onset or progression of periodontitis [41].

### 4.2. Physiopathology of Cardiovascular Diseases

The initial step in the development of atherosclerosis involves the accumulation and oxidation of Low Density Lipoprotein (LDL) particles in the subendothelial space, resulting from endothelial cell wall dysfunction. Oxidized LDL accumulates in the intima, leading to the activation, adhesion, and penetration of peripheral blood leukocytes through the endothelial wall. Monocytes differentiate into macrophages in the subendothelial space, where they capture and internalize large amounts of oxidized LDL, forming foam cells that are precursors of lipid streaks. Inflammatory cells, particularly macrophages, secrete various pro-inflammatory cytokines such as IL-1, TNF-alpha, and IL-6 [50]. These cytokines promote the recruitment of new leukocytes and induce the production of chemokines and cell adhesion molecules, sustaining chronic inflammation at the vascular level. Multiple biological pathways contribute to the development and progression of cardiovascular diseases, including atherosclerosis. Local cytokine release and inflammatory mediators cause structural changes in the vascular walls, recognized as factors in the development of atherosclerosis.

## 5. Dysbiosis of Oral Microbiota and Cardiometabolic Risk

Cardiometabolic diseases (CMDs) include a spectrum of conditions starting from insulin resistance and progressing to metabolic syndrome, pre-diabetes, cardiovascular diseases (CVDs), and type 2 diabetes (T2D) [51]. Several well-established risk factors contribute to CMDs, including increased waist circumference, inflammation indicated by high-sensitivity C-reactive protein (hsCRP) levels, hypertension, dysglycemia, dyslipidemia, decreased HDL levels, tobacco use, unhealthy diet, sedentary lifestyle, and psychosocial stress. Inflammation, characterized by the involvement of cytokines such as TNF-α, IL-1, and IL-6, plays a significant role in CMDs [52]. These diseases present a substantial global healthcare burden, with CVD being the leading cause of death worldwide and T2D affecting millions of individuals.

### 5.1. Pathophysiological Mechanisms Linking Periodontitis and CMDs

Dysbiosis of the oral microbiota is a known cause of periodontal disease and triggers an immuno-inflammatory reaction. Periodontal disease has been found to increase the risk of atherosclerosis, a condition characterized by inflammation and directly associated with cholesterol. Patients with cardiovascular disease have shown a 24% higher incidence of periodontal disease. Meta-analyses have demonstrated that appropriate periodontal treatment can reduce CMDs risk factors, improving plasma concentrations of inflammatory markers (C-reactive protein, interleukin-6, tumor necrosis factor-alpha), thrombotic markers (fibrinogen), and metabolic markers (triglycerides, total cholesterol, high-density lipoprotein cholesterol, glycated hemoglobin). However, more studies are needed to investigate the long-term effects of periodontal treatment on CMDs.

The scientific community agrees that chronic low-grade inflammation is the common etiopathogenic factor linking periodontal disease and CVDs and CMDs. This inflammation triggers the release of pro-inflammatory cytokines, leading to alterations in lipoproteins and their connection with associated receptors. Reduced receptor expression induced by inflammation impairs lipoprotein clearance, favoring a pro-atherogenic lipoprotein profile. Moreover, dysbiosis of the oral microbiota contributes to the decrease in anti-atherogenic processes. The association between periodontitis and imbalanced lipoprotein metabolism is particularly evident in lipoproteins containing B-100 apolipoprotein, including very low-density lipoproteins (VLDL), intermediate-density lipoproteins (IDL), and low-density lipoproteins (LDL).

### 5.2. Pathophysiology Linking Periodontitis and CMDs

The pathophysiology linking periodontitis and CMDs primarily involves bacteremia, endotoxemia, and low-grade systemic inflammation [24].

Bacteremia occurs when periodontal bacterial species invade the circulation through periodontal tissues during daily activities or professional interventions [41]. Periodontitis patients experience more frequent and longer episodes of bacteremia, involving more virulent bacterial species, compared to non-periodontitis patients [53]. Oral bacterial components, including DNA, RNA, and antigens from periodontal pathogens, have been detected in atherothrombotic tissues (such as Porphyromonadaceae). Animal models have shown that periodontal pathogens contribute to the increased incidence of CMDs risk factors and accelerate atherosclerosis [54]. Bacteremia can lead to bacterial colonization and growth on atherosclerotic coronary artery plaques, exacerbating coronary artery diseases [55].

Other periodontal pathogens have been found to directly invade various organs and tissues, including in the cardiovascular system. Bacteremia plays a crucial role in initiating endothelial lesions and exacerbating the vascular wall inflammatory process [52].

Endotoxemia, a consequence of periodontal bacteria release, is another significant mechanism. Lipopolysaccharides (LPS), major components of Gram-negative bacterial outer membranes, activate innate and adaptive immunity, triggering local and systemic inflammation. LPS stimulates the production of inflammatory mediators, cytokines, and matrix metalloproteinases, leading to periodontal tissue destruction [56]. LPS translocation into the bloodstream causes endotoxemia, impacting insulin levels and promoting insulin resistance, thereby increasing the risk of CMDs [57]. Chronic endotoxemia is involved in the pathogenesis of inflammatory conditions, including CMDs. Dysbiotic periodontal microbiota can contribute to endotoxemia in individuals with periodontitis [56]. Endotoxemia not only supports systemic inflammation, but also affects vessel walls and atherosclerotic lesions. It is positively correlated with triglyceride, cholesterol, and apolipoprotein B concentrations. High-fat diets can increase intestinal permeability, elevate LPS levels in circulation (metabolic endotoxemia), and raise the risk of CMDs and periodontitis [58]. Thus, endotoxemia serves as a molecular link between periodontitis and CMDs.

Low-grade systemic inflammation is another mechanism connecting periodontitis and extra-oral inflammatory comorbidities. Periodontitis triggers an immune-inflammatory response, resulting in the production of pro-inflammatory cytokines (TNF-alpha, IL-1beta, IL-6) that cause tissue damage [59]. These cytokines enter the bloodstream, inducing the release of C-reactive protein and the activation of cytokine networks, thereby promoting low-grade systemic inflammation [60]. Severe periodontitis patients exhibit elevated blood levels of pro-inflammatory mediators and increased neutrophil numbers, contributing to a state of inflammation throughout the body. Chronic systemic inflammation caused by periodontitis increases the risk of CVD and metabolic diseases. Additionally, the increase in adiposity and general inflammation resulting from metabolic diseases further exacerbates periodontal inflammation through NF-kB pathway activation [61]. These mechanisms are implicated in the formation of atherosclerotic lesions. Oxidative stress, which drives pro-inflammatory pathways common to CMDs and periodontitis, also represents a significant link between these pathologies [62]. Toll-like receptor 4 (TLR4) signaling plays a crucial role in pro-inflammatory signaling and the development of hypertension and diabetes, both contributing to CMDs. In fact, TLR4, the major receptor for LPS present in Gram-negative bacteria, highlights the potential role of oral microbiota and dysbiosis on hypertension. Chronic infections, including periodontal and peri-apical infections, may predispose individuals to higher risk of cardiovascular disease [63].

As explain in the figure below (Figure 1), there are some specific bacteria in a dysbiotic oral microbiota involved in low grade systemic inflammation that can lead to pathological heart, for example *Staphylococcus*, *Enterobacter*, *Ruminococcus*, etc.

### 5.3. Molecular Mechanisms of Bacterial Translocation Inducing Cardiometabolic Phenotypes

Finding the connection between dysbiosis of the oral microbiota and cardiometabolic phenotypes, such as insulin resistance, hepatic steatosis, and heart failure, involves identifying the molecular dialogue between the microbiota and the host. Initially, this involves bacterial molecules produced in the periodontal space that interact with the host’s epithelium and local immune system. One example is LPS, which can trigger local metabolic inflammation and also be transported by lipoproteins and plasma binding proteins, including LPS binding proteins released by adipose tissue [64]. The source of these circulating LPS is therefore associated with the increased permeability of the oral mucosa. During metabolic syndrome, both the oral and gut barriers become permeable through various mechanisms [65]. The contraction of tight junctions promotes their opening, allowing the passage of LPS multimers and other bacterial macromolecules. Transepithelial passage via M cells of the Peyer’s patches or mucus-secreting goblet cells has been observed [66]. In the latter case, translocated bacteria are typically efficiently eliminated by the local intestinal immune system. However, during type 2 diabetes induced by a high-fat diet and resulting dysbiosis, immune hypo-vigilance occurs, as observed in severely obese individuals undergoing bariatric surgery. Dysbiosis itself contributes to reduced immune competence. An analysis of antigen-presenting cells and helper lymphocyte transcriptomes reveals a decrease in bacterial recognition-related information transmission between these cell types [67]. Consequently, bacteria are no longer degraded locally by the intestinal immune system. Instead, these translocated bacteria are phagocytosed in the intestine and transported by phagocytes to metabolic tissues such as the liver, adipose deposits, heart, and even the brain. Following an excess fat-induced metabolic insult, the affected tissues generate chemokines that attract phagocytes, particularly those activated after bacterial phagocytosis. Phagocytes originate from both intestinal and periodontal sites, which are the primary locations of bacterial translocation.

Numerous studies have extensively explored the association between the oral microbiota and cardiometabolic diseases [68]. Epidemiologically, there is a clear positive correlation between periodontal disease and metabolic conditions. Patients with type 2 diabetes have a higher risk of developing periodontitis, while individuals with periodontitis have an increased likelihood of suffering from type 2 diabetes [69]. However, the exact molecular mechanisms responsible for this relationship are not yet fully understood. Dysbiosis of the oral microbiota leads to local inflammation, which contributes to the maintenance and exacerbation of a systemic metabolic inflammatory state throughout the body. This chronic inflammation can result in insulin resistance and various vascular and cardiometabolic disorders.

The connection between diabetes and periodontitis, which goes both ways, has been confirmed, with inflammation playing a role as a shared mediator. This positions periodontitis as the sixth complication of type 2 diabetes [70] and supports the notion that chronic low-grade inflammation [66], resulting from the dysbiosis of the oral microbiota, may be the driving force behind this pathology. Differences in the oral and periodontal microbiota have been observed in diabetic individuals compared to healthy individuals. Diabetic patients exhibit a significant increase in the abundance of genera such as *Aggregatibacter*, *Neisseria*, *Gemella*, *Eikenella*, *Selenomonas*, *Actinomyces*, *Capnocytophaga*, *Fusobacterium*, *Veillonella*, and *Streptococcus* [71]. Furthermore, the involvement of oral pathogens like *Porphyromonas gingivalis* (Pg) in insulin resistance has been demonstrated [58]. High levels of pro-inflammatory cytokines such as Tumor Necrosis Factor α (TNF-α) and Interleukin-6 (IL-6), produced by periodontal macrophages in response to Pg, increase the permeability of oral epithelial barriers [72]. This enhanced permeability promotes the translocation of Gram-negative bacteria and their virulence factors, including LPS, into the bloodstream. LPS not only induces an inflammatory state in organs, but also inhibits insulin receptor-mediated signaling pathways, thereby contributing to the development of insulin resistance [73]. Moreover, periodontal treatment has been shown to significantly reduce glycated hemoglobin (HbA1c) levels by up to 0.4% for patients with diabetes mellitus [74]. Additionally, the oral microbiota can influence the progression of diabetes, and certain oral hygiene practices, such as the excessive use of mouthwashes, may even have adverse effects on disease progression [75]. Nevertheless, many aspects of this complex interaction between CMDs and the oral microbiota still require further elucidation.

## 6. Treatment Strategies and Prevention

Oral Hygiene: The presence of oral bacterial species and their metabolites plays a significant role in the development and progression of periodontal disease and dental caries, including species like *Streptococcus mutans* and *Lactobacillus*. By effectively managing these acid-producing microorganisms through regular tooth brushing, the risk of caries and periodontal diseases can be minimized. Toothbrushing remains one of the simplest and most effective individual practices for maintaining good oral hygiene. Numerous studies have demonstrated that using fluoride toothpaste significantly reduces the prevalence of caries. Fluoride helps enhance enamel resistance to acidic pH levels and, when combined with arginine, contributes to maintaining a balanced oral microbial environment. While oral hygiene is crucial for sustaining a healthy oral microbiota, research exploring its impact on overall health remains limited. Nevertheless, one study revealed a positive association between tooth brushing, flossing, and reduced cardiovascular risk. More comprehensive research is required to precisely determine the role of oral hygiene in controlling cardiometabolic diseases.

Diet: Evidence suggests that dietary and lifestyle habits can influence oral diseases. The regular consumption of dairy products is inversely linked to the prevalence of periodontal disease. The nutrients, proteins, and probiotic bacteria found in dairy products are believed to have beneficial effects on periodontitis [76]. Furthermore, specific diets can have a positive impact on reducing the risk of heart disease and cardiovascular risk. As an example, the Mediterranean diet, known for its emphasis on consuming abundant fruits, vegetables, legumes, and olive oil, while reducing red meat intake and maintaining moderate wine consumption, is advised to prevent cardiovascular disease and type 2 diabetes.

Pre/probiotics treatment: As previously mentioned, *S. mutans* is recognized as the primary pathogen contributing to the development of caries. To counter its pathogenicity, the addition of probiotics like *L. acidophilus* has been shown to modify the balance, inhibiting pathogenic microorganisms while stimulating host defense mechanisms [77]. This approach holds promise in moving towards less invasive medical interventions. As the desire to combat the microbial resistance resulting from antibiotic treatments grows, the focus on neutralizing causative bacteria becomes increasingly important. The utilization of pre/probiotics to develop new prevention strategies becomes imperative. Similar methods have already been applied to the gut microbiota, employing specific probiotics like Bifidobacterium pseudocatenulatum and Bifidobacterium catenulatum, which are used to attenuated liver damage caused by reducing the impact of d-galactosamine. In mouse models, treatment with the probiotic Bifidobacterium pseudocatenulatum has demonstrated the ability to reduce obesity and inflammation by enhancing the epithelial barrier of the oral cavity.

Phytotherapy: In addition to the above approaches, alternative therapeutic strategies, such as herbal medicine, have emerged. One such example is puerarin, an active ingredient found in the root of pueraria lobata, which has been suggested to possess a potent anti-obesity effect. Puerarin treatment has been observed to increase the abundance of Akkermansia muciniphila, thereby safeguarding the intestinal barrier function by upregulating the expression of ZO-1 and occludin.

## 7. Conclusions

The primary challenge in managing CMDs revolves around their prevention and early detection, considering the individual variability in the development of these conditions. Therefore, it is crucial to comprehensively identify risk factors associated with CMDs, such as age, smoking, dyslipidemia, arterial hypertension, insulin resistance, and obesity. In this review, we strongly suggest the dysbiosis of the oral microbiota as a pivotal risk factor that needs to be understood and considered alongside other risk factors involved in CMDs. The specific goal of this review is to raise awareness among the medical community, urging them not to view dysbiotic oral microbiota associated with oral diseases as a deterministic causal factor, but rather as a stochastic risk factor in the context of CMDs. In other words, it is important to assess all risk factors and assign appropriate weights to develop a prevention and early diagnosis strategy. Furthermore, the concept of risk factors, thanks to emerging digital approaches, enables the development of technologies for predicting and diagnosing CMDs. This article also proposes innovative therapeutic strategies involving prebiotics, probiotics, and targeted bacterial treatments to modulate oral microbiota. Lastly, this narrative review places particular emphasis on the potential of maintaining oral hygiene, dietary choices, and phytotherapy as a means to prevent and manage cardiovascular diseases. Most notably, the research discussed in this article contributes to the growing field in leveraging oral microbiota as a novel risk factor and target for the prevention and treatment of cardiovascular diseases.

## Figures and Tables

**Figure 1 diagnostics-13-03184-f001:**
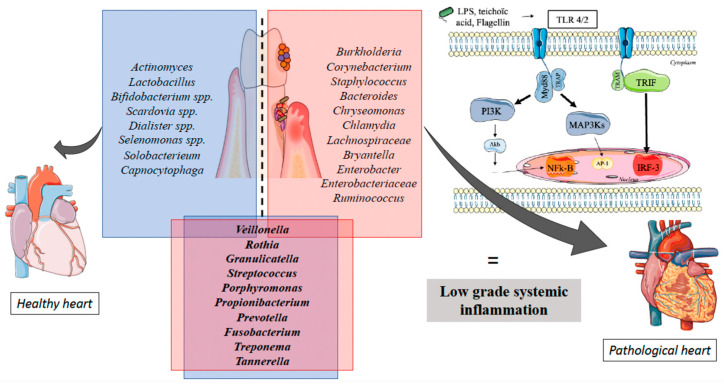
Summary of different bacteria and the mechanisms involved in low grade systemic inflammation leading to pathological heart.

**Table 1 diagnostics-13-03184-t001:** Main differences between salivary and periodontal microbiota.

	Salivary Microbiota	Periodontal Microbiota
Location	Circulating in the oral cavity	Space between gum and tooth (periodontium)
PH	3 < pH < 7	3 < pH < 8
Number of colony forming units (CFU)	10^9^ CFU/mL	10^11^ CFU/ML
Environment	Aerobic	Aero-anaerobic
Main bacterial family	Streptococcaceae	Fusobacteriaceae
	Neisseriaceae	Porphyromonadaceae
	Prevotellaceae	Prevotellaceae
	Staphylococcaceae	Tannerellaceae

**Table 2 diagnostics-13-03184-t002:** Involvement of bacterial families and species in cardiometabolic diseases (CMDs).

Family	Genus, Species	Features	Link with CMDs	Source
Streptococcaceae	*Streptococcus*, *Streptococcus mutans*	Gram +	- ↗ CM risk: ↗ HBP, ↘ HDL concentration - T2D - Endocarditis	- Fei N et al., PLoS ONE 2019 [26] - Karlsson FH et al., Nature, 2013.10/7/23 4:42:00 AM 10/7/23 4:42:00 AM [27]
Neisseriaceae	*Nesseiria*, *Neisseria* *flavescens*	Gram −	- ↘ HBP, ↗ vascular function and cardiometabolic outcomes	- E. Morou-Bermúdez et al., J Dent Res, 2022. [28]
Prevotellaceae	*Prevotella*, *Prevotella copri*	Gram −	- ↗ glucose tolerance	- Marungruang N et al., Eur J Nutr 2018. [29]
Staphylococcaceae	*Staphylococcus aureus*	Gram +	- ↗ Cholesterol, cardiometabolic alterations in children - Endocarditis	- Ayala-García JC et al., Metab Syndr Relat Disord 2022. [30]
Fusobacteriaceae	*Fusobacterium nucleatum*	Gram −	- Endocarditis - ↗ atherosclerosis	- Zhou J et al., Front Microbiol. 2022. [31]
Porphyromonada-ceae	*Porphymonas gingivalis*	Gram −	- ↗ NASH (Non-Alcoholic Steato-Hepatitis)	- Wang T et al., Front Med 2022. [15]
Tannerellaceae	*Tannerella forsythia*	Gram −	- ↗ Obesity - ↗ Hypertension, HBP	- Brandl B et al., Front Nutr. 2022. [32]

## Data Availability

Not applicable.

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
