# Peer review of "The Role of Dysbiotic Oral Microbiota in Cardiometabolic Diseases: A Narrative Review"

_diagnostics, 2023, doi:10.3390/diagnostics13203184_

Round 1

Reviewer 1 Report

In this review, the authors aimed to analyze the evidence for the association between the oral microbiota and cardiometabolic risk factors.
Please use justify alignment for the entire article.
Please specify in your article what type of review it is. Please add the methods of making the article.
Please specify the source of figure 1 or specify in which program it was made. Also, this figure seems unfinished...
I recommend you to have at least 2 figures and/or tables. Also a table summarizing the findings of different studies.
The article also has some mistakes from the technical editing point of view, which please correct them.
The article presents 70 references, being up to date. Reference 11 has no year.

Author Response

Thank you for giving us the opportunity to send you back a revised version of our manuscript “ Role of Dysbiotic Oral Microbiota in Cardiometabolic Diseases: a narrative review.”, diagnostics-2614668 B.

We have carefully studied the comments made by the two referees and have modified the text accordingly and included additional tables as they suggested.

Thus, we hope that you will now find our revised version suitable for publication in your journal.

Yours sincerely,

Reviewer 2 Report

The content of the submitted manuscript is good but the presentation way of the current form does not fulfill the journal requirements. Modification is needed to consider for publication.

 Please see the enclosed pdf for a point-by-point analysis.

English is fine

Author Response

(The authors gave the same response as above.)

Round 2

Reviewer 2 Report

The manuscript has been improved